# Multipotent Stromal Cell Extracellular Vesicle Distribution in Distant Organs after Introduction into a Bone Tissue Defect of a Limb

**DOI:** 10.3390/life11040306

**Published:** 2021-04-01

**Authors:** Igor Maiborodin, Aleksandr Shevela, Michael Toder, Sergey Marchukov, Natalya Tursunova, Marina Klinnikova, Vitalina Maiborodina, Elena Lushnikova, Andrew Shevela

**Affiliations:** 1Institute of Molecular Pathology and Pathomorphology, Federal State Budget Scientific Institution “Federal Research Center of Fundamental and Translational Medicine”, Ministry of Science and Higher Education of the Russian Federation, Akademika Timakova st., 2, 630117 Novosibirsk, Russia; natalya-tursunova@mail.ru (N.T.); margen@ngs.ru (M.K.); mai_@mail.ru (V.M.); pathol@inbox.ru (E.L.); 2The Center of New Medical Technologies, Institute of Chemical Biology and Fundamental Medicine, The Russian Academy of Sciences, Siberian Branch, Akademika Lavrenteva str., 8, 630090 Novosibirsk, Russia; mdshevela@gmail.com (A.S.); msv-1981@mail.ru (S.M.); ashevela@mail.ru (A.S.); 3International Center of Dental Implantology “iDent”, Sibrevkoma st., 9b, 630007 Novosibirsk, Russia; drtoder@gmail.com

**Keywords:** multipotent stromal cell extracellular vesicles, extracellular vesicles distribution, exosomes, bone tissue defect of a limb, lungs, heart, spleen, liver

## Abstract

When administered intravenously, extracellular vesicles derived from multipotent stromal cells (MSC EVs) immediately pass through the lungs along with the blood and regularly spread to all organs. When administered intraperitoneally, they are absorbed either into the blood or into the lymph and are quickly disseminated throughout the body. The possibility of generalized spread of MSC EVs to distant organs in case of local intratissular administration remains unexplored. However, it is impossible to exclude MSC EV influence on tissues distant from the injection site due to the active or passive migration of these injected nanoparticles through the vessels. The research is based on findings obtained when studying the samples of lungs, heart, spleen, and liver of outbred rabbits of both sexes weighing 3–4 kg at various times after the injection of EVs derived from MSCs of bone marrow origin and labeled by PKH26 into an artificially created defect of the proximal condyle of the tibia. MSC EVs were isolated by serial ultracentrifugation and characterized by transmission electron microscopy and flow cytometry. After the introduction of MSC EVs into the damaged proximal condyle of the tibia of rabbits, these MSC EVs can be found most frequently in the lungs, myocardium, liver, and spleen. MSC EVs enter all of these organs with the blood flow. The lungs contained the maximum number of labeled MSC EVs; moreover, they were often associated with detritus and were located in the lumen of the alveoli. In the capillary network of various organs except the myocardium, MSC EVs are adsorbed by paravasal phagocytes; in some cases, specifically labeled small dust-like objects can be detected throughout the entire experiment—up to ten days of observation. Therefore, we can conclude that the entire body, including distant organs, is effected both by antigenic detritus, which appeared in the bloodstream after extensive surgery, and MSC EVs introduced from the outside.

## 1. Introduction

The fact that multipotent stromal cells (MSCs) injected in the tissue are disseminated throughout the body via the blood flow can be considered proven to-date [1,2,3,4]. Several years ago, angiogenesis involving MSCs that migrated through the lymphatic vessels was shown at a distance from the injection site [5]. Due to the short life of MSCs in tissues [6,7,8,9,10], the focus of cellular research has gradually shifted from MSCs themselves to their extracellular vesicles (MSC EVs). However, it is also impossible to exclude MSC EV influence on tissues distant from the injection site due to the active or passive migration of these injected nanoparticles through the vessels. In this regard, it is relevant to study the possibility of MSC EV bio-distribution throughout the body after a local administration.

It should be noted that similar research was conducted by other researchers. Nanovesicles were obtained from MSCs by isolation in a density gradient as a result of cell disruption by successive extrusions. These vesicles, labeled with Cy7 (cyanine fluorescent dye), were found in the lungs, liver, and kidneys six hours after intraperitoneal administration to mice at a dose of 2 × 10^9^. It was concluded that such objects are distributed throughout the animal’s organism [11].

Labeled extracellular vesicles from human MSCs were injected intravenously into mice with glycerol-induced kidney damage as well as into healthy animals. The labeled vesicles were found to accumulate in damaged kidneys compared with healthy controls. After five hours, such vesicles were found in full body images and kidney sections [12].

However, when administered intravenously, MSC EVs immediately pass through the lungs along with the blood and regularly spread to all organs. When administered intraperitoneally, they are absorbed either into the blood or into the lymph and are quickly disseminated throughout the body. The possibility of generalized spread of MSC EVs to distant organs in case of local intratissular administration remains unexplored.

In connection with the above, the aim of the study was formulated as follows: Searching for MSC EVs labeled with a fluorescent dye in the distant organs after their injection into an artificially created bone tissue defect of a limb using the methods of luminescence microscopy.

## 2. Materials and Methods

The research is based on findings obtained when studying the samples of lungs, heart, spleen, and liver of outbred rabbits of both sexes weighing 3–4 kg at various times after the injection of MSC EVs into an artificially created defect of the proximal condyle of the tibia with subsequent installation of screw titanium implants. The manipulations did not cause pain to animals and were carried out in compliance with Russian legislation: GOST 33215-2014 (Guidelines for accommodation and care of laboratory animals. Rules for equipment of premises and organization of procedures) and GOST 33216-2014 (Guidelines for accommodation and care of laboratory animals. Rules for the accommodation and care of laboratory rodents and rabbits). Work approved by the Committee on Biomedical Ethics (20 November 2020, decision No. 32) at the Federal State Budget Scientific Institution Federal Research Center of Fundamental and Translational Medicine (FRC FTM).

### 2.1. Preparation, Cultivation, and Characteristics of MSCs; Isolation of MSC EVs

Due to the numerous literature publications that undifferentiated MSCs do not have surface antigens that determine their foreignness, do not initiate an immune system response even after allogenic and xenogenic transplantation, and the inflammation and proliferation of allogeneic T-lymphocytes is also suppressed, it is possible to use MSCs isolated from some animals for implantation in other individuals, even another species [13,14]. Similar data were obtained in the study of the immunogenicity of EVs derived from human umbilical cord MSCs. EVs were given to rabbits, guinea pigs, and rats, and EVs have been found to have a protective effect on weight loss and had no adverse effects on liver or renal function. Other detections such as hemolysis, vascular and muscle stimulation, systemic anaphylaxis, pyrogen, and hematology indexes also showed exosomes were applicable. Thus EVs from human umbilical cord MSCs are well tolerated in animal models [15]. Moreover, there is evidence of the possibility of using EVs of plant origin for the treatment of various pathological processes in mammals. Currently, there is evidence that EVs of plant origin may be involved not only in plant–cell communication but also in interspecies communication between plants and animals. For example, a plant-derived miRNA such as miR-168 has been reported to enter the circulation of rice-fed mice enclosed in EVs and to modulate the expression of target genes [16]. EVs are released from microorganisms and may participate in interspecies communication in the gut [17]. Based on the above, a decision was made on the possibility of transplanting EVs obtained from rat MSCs into rabbits.

MSCs were obtained from the bone marrow of a male WAG inbred line rat weighing 180 g and aged 6 months and then were characterized and cultured as described in our previous works [3,4,5,7,8,10]. Isolated MSCs expressed some characteristic MSC markers (CD73, CD90, and CD105) and did not express the hematopoietic markers (CD14, CD20, CD45, and CD34). The culture nutrient medium containing fetal bovine serum and used for MSC growth was preliminarily subjected to complete gradient centrifugation and purified from its own EVs. At the stage of stationary growth of a stable culture of the 3rd MSC passage, when the confluence of the cell monolayer reached 80–90%, a conditioned medium was collected from which MSC EVs were isolated as recommended in literature [18,19]. To remove cells, cell debris, apoptotic bodies, and large vesicles, the conditioned medium was centrifuged sequentially: 10 min in case of 300 *g*, 10 min in case of 2000 *g*, and 30 min in case of 12,000 *g*. EVs were precipitated by centrifuging the supernatant for 2 h in case of 100,000 *g* and resuspended in saline with phosphate buffer. MSC EVs were analyzed by electron transmission microscopy and flow cytometry. Relevant markers for EVs secreting MSCs, tetraspanins CD9, CD63, and CD81, were detectable [20,21]. When preparing samples for research, the objects were sorbed onto a copper grid covered with a formvar film for 1 min and contrasted with a 2% phosphotungstic acid solution for 10 s. The grids were studied in the transmission mode of a Jem1400 electron microscope (Jeol, Japan); images were obtained using a Veleta digital camera (Olympus Corporation, Japan). Particle size was determined in 5–8 randomly selected fields of view at 60,000 times magnification using the iTEM software package (Olympus Corporation, Japan). More than 90% of the objects had a diameter of 70–90 microns and a three-layer membrane. The isolated extracellular particles were adsorbed onto 4 μm aldehyde/sulfate latex particles (Invitrogen, Waltham, MA, USA, 37304) [22] and analyzed on a NovoCyte ™ cytometer using its software (ACEA Biosciences Inc., San Diego, CA, USA). To detect EVs, antibodies specific to marker-specific exosome proteins (Bio-Rad, Hercules, CA, USA, MCA4754F, FITC mouse IgG1, k) were used. An isotype control (Bio-Rad, MCA1209, isotype control, FITC mouse IgG1, k) was used to measure nonspecific sorption. In each experiment, no less than 30,000 events were counted. The amount of MSC EVs was determined by the protein content in the precipitate using a commercial Qubit protein assay kit (Thermo Fisher Scientific, Waltham, MA, USA) and a Qubit^®^ 3.0 fluorometer.

The lipid components of exosome membranes were stained with PKH26 red fluorescent dye according to the manufacturer’s instructions (Sigma-Aldrich, St. Louis, MO, USA). The excess dye was removed by centrifugation for 10 min in case of 2000 *g* and for 2 h in case of 100,000 *g* or using Exosome Spin Columns (MW 3000) following a standard protocol (Thermo Fisher Scientific).

### 2.2. Introduction of MSC EVs into a Bone Defect

Surgical intervention was performed in compliance with all the rules of asepsis and antiseptics under general intravenous anesthesia with propofol. In both proximal tibial condyles of rabbits, standardized 4 mm holes were created with a 2 mm dental bur and cooled by sterile saline solution [19].

Next, an insulin syringe was used to fill the bone defect with physiological saline prepared in phosphate buffer (pH = 7.3) (control, 9 rabbits) [23] or 19.2 μg of MSC EVs in saline solution injected for each limb (experiment, 10 animals). The MSC EVs dose was selected based on the average dose recommended by other researchers: 10–20 μg/mL [24]; 0.6 μg, 5 μg, and 50 μg [25]; 50 μg for the same bone tissue defect of the proximal tibial condyle [19]; 100 μg immediately after surgery and weekly for 12 weeks [23]. After 10–20 s, titanium screw implants (catalog number IS 358; 3.5 × 8 mm with a rough surface; 3S, Israel) were inserted with a stable primary fixation up to 30 Ncm, and the surgical wound was sutured layer by layer without tension.

After 3, 7, and 10 days, the animals were sacrificed by dislocation of the cervical vertebrae. Each group consisted of 3–4 animals, 19 animals in total.

### 2.3. Morphological Research Methods

Samples of the diaphragmatic lobe of the right lung, left myocardium, central region of the spleen, and left external lobe of the liver were fixed in 4% paraformaldehyde solution in phosphate buffer (pH 7.4) for at least 1 day, then dehydrated and clarified in the Isoprep reagent (BioVitrum, Saint Petersburg, Russia) and enclosed in a histoplast.

Unstained sections were examined in the luminescence mode of Axioimager M1 microscope using color filters for Alexa Fluor 488 (excitation range 450–490 nm, registration range 515–∞ nm) and for rhodamine (Rhod excitation range 540–552 nm, registration range 575–640 nm). Tissue examination under UV light with Alexa Fluor 488 filter was aimed for detecting green background autofluorescence which, firstly, provides good contrast for other luminescent objects, e.g., with rhodamine filter (red color and its shades are clearly visible on a green background). Secondly, it is possible to see the structure of the studied tissues and better localize the objects with red luminescence in a distinct organ [3,4,10,18,26].

Automatic exposure was used for obtaining microphotographs when combining the images using Alexa Fluor 488 and rhodamine filters. Thus, one may obtain green and red (or orange and yellow) color, depending on prevalence of the glow intensity using distinct filters. Brighter fluorescence when using an Alexa Fluor 488 filter gave green luminescence, while using a rhodamine filter produced red color. Yellow luminescence and its shades resulted from mixing green and red colors at different ratios [3,4,10,18,26].

## 3. Research Findings and Their Discussion

In all the examined organs of the control animals, objects with a predominant fluorescence in the time of using a rhodamine filter were not found at all points of the experiment.

On day three after the injection of MSC EVs, numerous objects of very small size, almost dust-like, with bright fluorescence were found when using the rhodamine filter in the ***lungs*** of all four rabbits. It is noteworthy that all such red glowing particles were located in the alveoli (Figure 1a).

Upon careful study at high magnification, it was noted that such fluorescent objects were sometimes freely located in the lumen of the alveoli (Figure 1b) while in other cases-, in some relatively homogeneous structures (Figure 1b,c), they were sometimes very large, reaching 20 μm in diameter (Figure 1c). At the level of luminescence microscopy, such volumetric formations did not show any cellular components but they contained either a few (Figure 1b) or very many (Figure 1c) small objects less than 1 μm in size with intense luminescence when a rhodamine filter was installed.

Most likely, such fluorescent red fine objects in the lungs were the MSC EVs, the membrane structures of which were stained with the luminescent dye PKH26. When MSC EVs were injected into an artificially created defect in the proximal tibial condyle before the implantation of a screw metal implant, a part of the MSC EVs entered the soft tissues, and another part was squeezed there when the foreign body was twisted into the bone tissue.

Both soft tissues on the surface of the condyle and the bone tissue itself, especially the red bone marrow, contain many blood vessels that are damaged both during the preparation and implantation. Therefore, it is possible for MSC EVs to enter the blood and be transported with the blood flow through the right parts of the heart into the pulmonary circulation—into the lungs where MSC EVs, possibly as fine foreign particles, leave the vessels passing through the alveolar septa and end up in the lumen of the alveoli.

In addition to MSC EVs, cellular and tissue debris including fibrin clots, which appear during the resorption of hemorrhages in damaged tissues, enter the lungs from the surgical site through the blood flow. When using MSC EVs immediately after surgery, their merging is possible due to fibrin from hemorrhages as well as adhesion to fragments of tissue detritus and phagocytosis, with detritus and fibrin by immunocompetent cells. Next, detritus with MSC EVs, phagocytes with MSC EVs, and fibrin clots with MSC EVs enter the blood flow through the damaged vessels and end up in the right heart cavities. Then, all this migrates into the lungs with the blood flow where, after passing through the vessel membranes, it appears in the lumen alveoli. Most likely, large homogeneous structures with characteristic point luminescence are either fibrin clots, fragments of detritus, or dead macrophages (or groups of phagocytes) that entered the lungs together with adhered or adsorbed MSC EVs from the site of surgery or intratissular injection of these MSC EVs.

The reality of these assumptions is evidenced by very large, more than 20 μm, cells (which were found in two animals) of various forms located in the walls of the pulmonary vessels or directly next to them and having a clear red tint of fluorescence of numerous different-sized cytoplasmic inclusions (Figure 1d). Most likely, these large cells are paravascular macrophages, and the luminescence of cytoplasmic inclusions (phagosomes) is associated with phagocytosis of detritus together with labeled MSC EVs from the blood flow when a rhodamine filter is used. The possibility that macrophages located near the vessels acquired the ability to fluorescence due to phagocytosis from the bloodstream of luminescent objects was previously shown in the literature [7,8,10].

On days seven and ten, single, very rare objects that glowed when we applied the rhodamine filter were found in only one out of three animals within each period. These objects were still located in the alveoli and were no longer dust-like, but had a size of about 5–7 μm. The fluorescence intensity decreased slightly; sometimes it was not a pure red color, but rather orange (Figure 1e,f).

Apparently, by this time, pure MSC EVs no longer entered the bloodstream and, accordingly, the lungs, but fragments of detritus or immunocompetent cells with MSC EVs could still be transported. Gradually, the activity of inflammation in the condyle damaged during surgery decreased, along with the volume of detritus that entered the blood and the lungs together with MSC EVs. In addition, in comparison with the previous period, the number of MSC EVs introduced during surgery also decreased in the tissues, which means that their volume should be smaller both in detritus and in immunocompetent cells participating in the elimination of MSC EVs introduced from outside.

Three days after surgery using MSC EVs, very small individual objects with intense fluorescence using a rhodamine filter were found in the ***myocardium***. Such objects have always been located in the vascular structures of the heart muscle and less often on the endothelial lining of valvular formations or the atrium or ventricle proper. None of the four animals showed tissue detritus or cells with red glowing objects during observations (Figure 2a).

From the injection site, MSC EVs enter the right heart either on their own or with detritus, or in cells transported by the blood flow. However, the myocardium is supplied with the blood through the coronary arteries extending from the aorta, where blood is transported from the lungs together with MSC EVs. That is, from the right atrium and ventricle of the heart, MSC EVs cannot enter the myocardium but can remain on the endothelium of these cavities.

Once in the capillaries of the lungs, detritus with MSC EVs and cells with MSC EVs, especially large phagocytes, are mostly filtered out and absorbed by the macrophages of the lungs or are eliminated into the alveoli and further outward as shown above when describing the lungs (Figure 1b–d), as well as detritus of MSCs also labeled with the luminescent dye [3,4]. And, most likely, this is why there were only single small, dust-like objects with a very bright red glow when a rhodamine filter was installed in the heart muscle that is, MSC EVs. The location of MSC EVs in the lumen of capillaries, on valves, and chords, as well as in the vessel wall, is additional evidence that such objects with a red glow are not artifacts, but have entered the heart through vessels with the blood flow. Due to a high pressure in the cardiac vessels as well as the contractile activity of the myocardium, MSC EVs move through the vessels deep into the heart muscle, getting into its capillaries.

After seven days, the picture in the myocardium was practically unchanged. As before, the vascular structures of the heart muscle contained very dust-like objects with red fluorescence when the rhodamine filter was used (Figure 2b).

By the end of the observation on day ten, only small objects that glowed weakly when using the rhodamine filter could be found in the myocardium, which were extremely rare and were found only in one rabbit out of three as a result of a careful search. Given the gradual subsiding of inflammation in the injured tissues of the hind limb as well as the corresponding decrease in the number of objects and the intensity of red fluorescence in the lungs, it can be assumed that almost complete disappearance of brightly luminescent structures from the heart is due to the cessation of the detritus flow with MSC EVs into the blood from the site of surgery. By the same time, MSC EVs that entered the heart muscle before are eliminated: they are either decayed or adsorbed by certain cells.

On the third day after surgery using MSC EVs, a dust-like object with a clear, undoubted predominance of fluorescence when applying the rhodamine filter was found in the ***spleen*** only in one case out of four. This object was visually close to a large macrophage with strong autofluorescence located in the red pulp of the organ and, possibly, was adhered to the surface of the phagocyte or was even in the process of being absorbed (Figure 2c).

In all other observations on this day and for the rest of the days, similar objects were not found (Figure 2d). At the same time, a pronounced red tint in the glow of numerous macrophages of the red and white pulp of the organ should be noted. Such phagocytes were not very common, but were found in the spleen of each animal at each observation period. This tint was not in the entire cytoplasm of phagocytes and not in all luminous cytoplasmic inclusions, but only in certain clearly limited structures, i.e., lysosomes (Figure 2d).

The initial pronounced autofluorescence of spleen phagocytes should be taken into account as well as the fact that splenic macrophages (siderophages) participate in the utilization of damaged and old erythrocytes, while the luminescence of phagocytes can be provided by the adsorption of fluorescent substances [7,8,10], erythrocytes, and their degradation products [27,28].

As shown above, the injected MSC EVs enter the lungs from the tibial defect, pass, most likely, partially, by the pulmonary macrophages, and end up in the heart muscle. Since the spleen is supplied with blood through a rather thick splenic artery (a truncus celiacus branch extending from the abdominal aorta), MSC EVs, which have passed through the vessels of the lungs, must also enter this organ.

It can be assumed that MSC EVs actually end up in the spleen, as evidenced by a red-glowing dust-like object (Figure 2c), but once there, they are immediately phagocytized by macrophages which are present in the organ in large quantities. Since, in addition to MSC EVs, spleen macrophages adsorb many other substances, some of which have the ability to autofluorescence (erythrocytes and hemosiderin [27,28]), the luminescence of macrophages due to MSC EVs labeled with PKH26 can be masked by the luminescence of other objects also found in phagosomes.

Within a period of three days after surgery using MSC EVs, macrophages with a red tint in the fluorescence of individual inclusions were found in the ***liver*** of each animal. These macrophages were located along the sinusoids; it was impossible to find a special tropism in the periphery vessels or veins in the center of the lobules (Figure 2e). In addition, very small objects with predominant luminescence were found in two out of four rabbits directly in the liver parenchyma when the rhodamine filter was installed.

That is, MSC EVs from the injection site located in the hind limb tissues could be found in the liver of rabbits on day three. These MSC EVs must have passed through the capillary network of the lungs before entering the liver. Most likely, MSC EVs were transported to the liver with the blood through the hepatic artery system, and then MSC EVs in the sinusoids either adhered to the endothelium, were phagocytosed by Kupffer’s cells, or were adsorbed by hepatocytes or other cells of this organ. As noted above, the ability of paravascular macrophages to adsorb foreign substances labeled with a luminescent dye from the bloodstream was reported in literature, as well as the ability of phagocytes to fluoresce noticeably due to the accumulated label [7,8,10].

By the seventh day after surgery using MSC EVs, many macrophages with a clear red tint of luminescence were noted in the liver when a rhodamine filter was installed. Along with this, small, dust-like objects disappeared from the vascular system and the organ parenchyma (Figure 2f).

After surgery, detritus with MSC EVs enters the bloodstream from the damaged bones of the hind limbs. Debris and part of the MSC EVs are adsorbed by the macrophages of the lungs, and the remaining MSC EVs are distributed throughout the body via the systemic circulation system. Apparently, by the seventh day, the flow of MSC EVs from the surgical site into the blood stops or decreases significantly due to the repair of damaged tissues. Due to this, the MSC EVs—very small dust-like objects with a bright glow when a rhodamine filter is installed—disappear from the liver while macrophages remain, which earlier adsorbed the MSC EVs from the blood flow and, as a result, acquired the ability to glow red due to the presence of fragments of MSC EVs with PKH26-labeled membranes in phagosomes or PKH26 staining of lysosomal and other membranes of the phagocytes themselves.

Since the degradation of MSC EVs occurs gradually in macrophages, and phagocytes adsorb not only MSC EVs but also many other objects from the blood, the intensity of the red luminescence of phagosomes in macrophages is much less pronounced than that of the MSC EVs and has only a red tint in some cytoplasmic inclusions. By the tenth day, objects with predominant or even weak luminescence when using the rhodamine filter were not found in the liver. 

By this time, the alterative phase of inflammation in the hind limb tissues damaged during surgery ends. Accordingly, the flow of detritus and MSC EVs into the blood and their dissemination throughout the body after the filtration in the lungs are significantly reduced. Gradually, phagocytes with MSC EVs or their fragments in lysosomes either migrate from the liver or the fluorescent dye concentration decreases to a level that is invisible during visual observation. 

## 4. Conclusions

Thus, after the injection of MSC EVs into the damaged proximal condyle of the tibia of rabbits, these MSC EVs can be found most frequently in the lungs, myocardium, liver, and spleen. MSC EVs enter all of these organs with the blood flow. The lungs contained the maximum number of labeled MSC EVs; moreover, they were often associated with detritus and were located in the lumen of the alveoli. That is, from the site of local administration, a part of the MSC EVs was transported to the right heart with the blood flow and then to the lungs. After filtration in the capillaries of the alveolar septa of detritus with MSC EVs and, to some extent, free MSC EVs which can be eliminated into the lumen of the alveoli, the remaining vesicles were distributed with the blood flow throughout the body. In the capillary network of various organs except the myocardium, MSC EVs are adsorbed by paravasal phagocytes; in some cases, specifically labeled small, dust-like objects can be detected throughout the entire experiment - up to ten days of observation. Therefore, we can conclude that the entire body, including distant organs, can be affected both by antigenic detritus which appeared in the bloodstream after extensive surgery, and MSC EVs introduced from the outside.

## Figures and Tables

**Figure 1 life-11-00306-f001:**
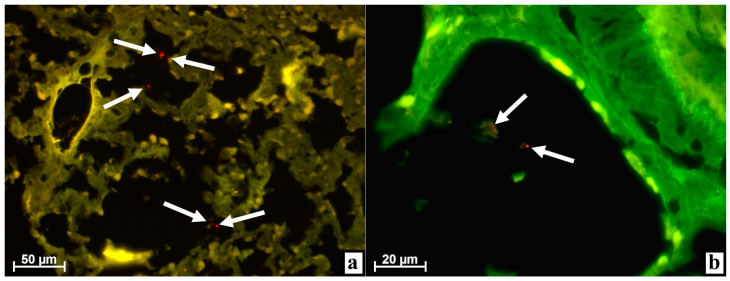
Lungs of rabbits at various times after injecting extracellular vesicles derived from multipotent stromal cells (MSC EVs) into the tibial defect. Alignment of images obtained using Alexa 488 and rhodamine filters in the luminescent mode of the microscope. (**a**) After 3 days, numerous objects of very small size, dust-like, with bright fluorescence when using the rhodamine filter (arrows) were found in the alveoli. (**b**) In the alveolar lumen, objects with luminescence when applying the rhodamine filter were associated with a structureless substance (arrows) on day 3. (**c**) After 3 days, numerous objects with very bright fluorescence when using the rhodamine filter were enclosed in a homogeneous substance in the alveolus of the lung (arrows). (**d**) Cytoplasmic inclusions in large cells of various shapes have a red glow when applying the rhodamine filter in the lung parenchyma near a large vessel (the lumen is indicated by an arrow) on day 3. (**e**) By the 7th day, an object of about 5 μm in size (arrow) with intense fluorescence when using the rhodamine filter was found in the alveolus. (**f**) After 10 days, the alveolus contained an object with a diameter of 5–7 μm (arrow) with a very bright glow when a rhodamine filter was installed.

**Figure 2 life-11-00306-f002:**
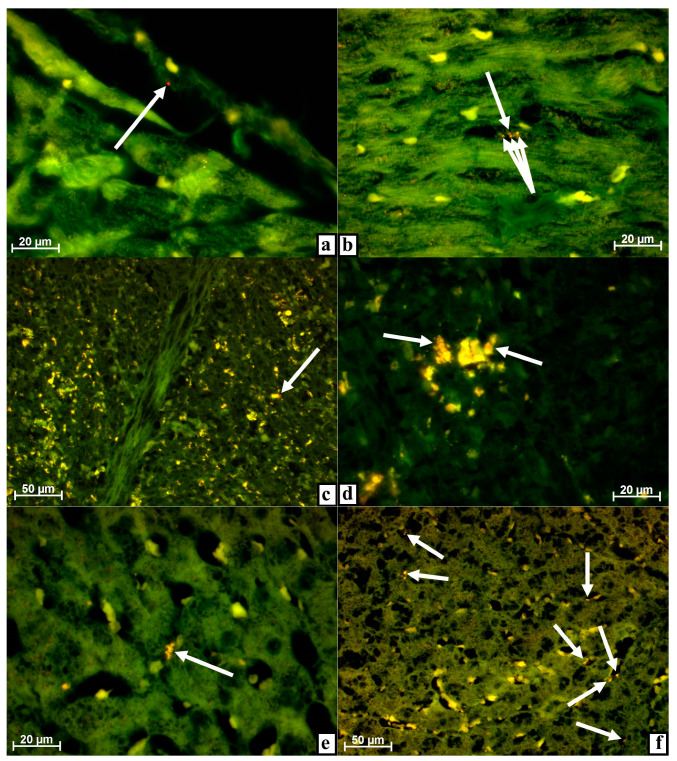
Myocardium (**a,b**), spleen (**c,d**), and liver (**e,f**) of rabbits at different dates after damage to the tibia with the introduction of MSC EVs. Alignment of images obtained using Alexa 488 and rhodamine filters in the luminescent mode of the microscope. (**a**) On day 3, a very fine, dust-like object with strong fluorescence was located on the capillary endothelium (arrow) when a rhodamine filter was applied. (**b**) After 7 days, the capillary contained several very small, dust-like objects with a bright glow (arrow) when a rhodamine filter was installed. (**c**) After 3 days, a very small, dust-like object with a strong glow was noticeable close to a macrophage with intense autofluorescence located in the red pulp (arrow) when a rhodamine filter was used. (**d**) Individual inclusions in the cytoplasm of red pulp macrophages fluoresce with a noticeable red tint (arrows) by day 7 when a rhodamine filter was installed. (**e**) After 3 days, a macrophage in a sinusoid had a red tint in the glow of cytoplasmic inclusions (arrows) when a rhodamine filter was applied. (**f**) After 7 days, numerous macrophages were located along the sinusoids fluoresce with a clear red tint (arrows) when a rhodamine filter was installed.

## Data Availability

The data presented in this study are available on request from the corresponding author.

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
