# Peer review of "Multipotent Stromal Cell Extracellular Vesicle Distribution in Distant Organs after Introduction into a Bone Tissue Defect of a Limb"

_life, 2021, doi:10.3390/life11040306_

Round 1
Reviewer 1 Report
The lack of clarity and fluency makes the manuscript, “Multipotent Stromal Cell Extracellular Microvesicle Distribution in Distant Organs after Introduction Into a Bone Tissue Defect of a Limb” too hard to understand and follow. I could not clearly understand it to provide the scientific comments. The scientific review could be possible after revising the manuscript to be clear and fluent enough to read.
Author Response
I am very sorry that the reviewer was unable to read our manuscript. I apologize, but I find it necessary to point out that the translation and verification of the text was done by a certified manuscript translation agency.
The translation made by the translation agency is attached.

Reviewer 2 Report
In their work, Maiborodin I, et al. investigated biodistribution of multipotent stromal cells-derived extracellular microvesicles (MSC-MVs) in selected organs of the experimental animals (including lungs, heart, liver and spleen), after local administration into the artificially created defect of the tibia. The Authors provide novel and important data, which are comprehensively discussed. Regarding rapidly expanding extracellular vesicles-related research field, and the potential use of MSC-MVs in regenerative medicine, the findings from this article may attract a broad audience of readers. Although interesting, this work has several limitations, which are listed below:
Major points:
- The Authors use the term “microvesicles”, however, according to the guidelines by the International Society for Extracellular Vesicles (ISEV) launched in 2018 (MISEV2018; doi: 10.1080/20013078.2018.1535750), if not experimentally defined, the general term which should be used, is “extracellular vesicles” (EVs).
- The studies of EVs constitute the primary goal of this work. However, no characterization or visualization of the EVs is provided by any method. To follow the MISEV2018 criteria, the Authors must show EVs either by TEM or AFM and detect typical EV-markers using, e.g. Western blot method.
- Why have the Authors use rat MSC-EV in a rabbit transplantation model? Are there any inter-species immune barriers that should be overcome? Please, discuss.
- Did the Authors use FBS-containing medium to collect MSC-EVs? The “Materials and methods” section does not clearly describe the procedure. In case if MSC-EVs were collected in serum-supplemented cell culture medium, additional controls should be used to prove that truly MSC-MVs (not FBS-EVs) were analyzed in the study.
- The information abut PKH26 dye in the “Materials and methods” section is incomplete. Although it can be concluded from the results and the discussion part that red fluorescence was used, such information should be included in the materials section.
Minor points:
- In the introduction, the Authors cite two studies utilizing fluorescently labeled EVs (ref 11 and 12). However, MSC may originate from different tissues (bone marrow, adipose tissue, umbilical cord, etc.), which may affect EV properties. Therefore, it should be stated which population of MSCs was exactly used in these studies.
- The “Author contributions” section is not completed. Please, complete.
- The “Acknowledgements” are incorrect. Please, provide correct version.
Author Response
I thank the reviewer for studying our manuscript and for the comments made
Major points:
- The Authors use the term “microvesicles”, however, according to the guidelines by the International Society for Extracellular Vesicles (ISEV) launched in 2018 (MISEV2018; doi: 10.1080/20013078.2018.1535750), if not experimentally defined, the general term which should be used, is “extracellular vesicles” (EVs).
Changed
- The studies of EVs constitute the primary goal of this work. However, no characterization or visualization of the EVs is provided by any method. To follow the MISEV2018 criteria, the Authors must show EVs either by TEM or AFM and detect typical EV-markers using, e.g. Western blot method.
EVs were analyzed by electron transmission microscopy and flow cytometry .Relevant marker for EVs secreting MSCs, tetraspanins CD9, CD63 and CD81 were detectable (Lötvall J, Hill AF, Hochberg F, Buzás EI, Di Vizio D, Gardiner C, Gho YS, Kurochkin IV, Mathivanan S, Quesenberry P, Sahoo S, Tahara H, Wauben MH, Witwer KW, Théry C. Minimal experimental requirements for definition of extracellular vesicles and their functions: a position statement from the International Society for Extracellular Vesicles. J Extracell Vesicles. 2014 Dec 22;3:26913. doi: 10.3402/jev.v3.26913. PMID: 25536934; PMCID: PMC4275645.; Witwer KW, Soekmadji C, Hill AF, Wauben MH, Buzás EI, Di Vizio D, Falcon-Perez JM, Gardiner C, Hochberg F, Kurochkin IV, Lötvall J, Mathivanan S, Nieuwland R, Sahoo S, Tahara H, Torrecilhas AC, Weaver AM, Yin H, Zheng L, Gho YS, Quesenberry P, Théry C. Updating the MISEV minimal requirements for extracellular vesicle studies: building bridges to reproducibility. J Extracell Vesicles. 2017 Nov 15;6(1):1396823. doi: 10.1080/20013078.2017.1396823. PMID: 29184626; PMCID: PMC5698937.).
When preparing samples for research, the objects were sorbed onto a copper grid covered with a formvar film for 1 min and contrasted with a 2% phosphotungstic acid solution for 10 sec. The grids were studied in the transmission mode of a Jem1400 electron microscope (Jeol, Japan); images were obtained using a Veleta digital camera (Olympus Corporation, Japan). Particle size was determined in 5-8 randomly selected fields of view at 60 000 times magnification using the iTEM software package (Olympus Corporation). More than 90% of the objects had a diameter of 70-90 microns, Japan.
The isolated extracellular particles were adsorbed onto 4 μm aldehyde/sulfate latex particles (Invitrogen, 37304) (Dias MV, Martins VR, Hajj GN. Stress-Inducible Protein 1 (STI1): Extracellular Vesicle Analysis and Quantification. Methods Mol Biol. 2016;1459:161-74. doi: 10.1007/978-1-4939-3804-9_11.) and analyzed on a NovoCyte ™ cytometer using its software (ACEA Biosciences Inc.). To detect EV, antibodies specific to marker-specific exosome proteins (Bio-Rad, MCA4754F, FITC mouse IgG1, k) were used. An isotype control (Bio-Rad, MCA1209, isotype control, FITC mouse IgG1, k) was used to measure nonspecific sorption. In each experiment, no less than 30 000 events were counted.
The text of the manuscript is supplemented
- Why have the Authors use rat MSC-EV in a rabbit transplantation model? Are there any inter-species immune barriers that should be overcome? Please, discuss.
Due to the numerous literature publications that undifferentiated MSCs do not have surface antigens that determine their foreignness, do not initiate an immune system response even after allogeneic and xenogenic transplantation, the inflammation and proliferation of allogeneic T-lymphocytes is also suppressed, it is possible to use MSCs isolated from some animals, for implantation in other individuals, even another species (Caplan AI, 2009; Poncelet AJ et al., 2009; Undale AH et al., 2009; Yamaza T. et al., 2010; Charbord P., 2010).
Similar data were obtained in the study of the immunogenicity of exosomes. The safety of transplantation of exosomes derived from human umbilical cord MSCs has been studied. Exosomes were incubated with the cardiac blood from a healthy rabbit, and hemolysis was observed. For analysis of vascular and muscle stimulation, pyrogen, systemic anaphylaxis and hematology indexes, exosomes were given to rabbits, guinea pigs and rats. The histological changes in the vascular and muscle sites of injection in rabbits were analyzed by hematoxylin and eosin staining. Allergy symptoms in guinea pigs and rectal temperature of rabbits were observed and recorded. To study safety in vivo, exosomes were infused intravenously into rats with acute myocardial infarction. Rats' weight was measured and tail vein blood was collected to evaluate liver and renal function. Exosomes have been found to have a protective effect on weight loss and had no adverse effects on liver or renal function. Other detections, such as hemolysis, vascular and muscle stimulation, systemic anaphylaxis, pyrogen and hematology indexes, also showed exosomes were applicable. Thus exosomes from human umbilical cord MSCs are well tolerated in animal models. This study provides evidence for the safety of intravenous infusion in future clinical therapy (Sun L, Xu R, Sun X, Duan Y, Han Y, Zhao Y, Qian H, Zhu W, Xu W. Safety evaluation of exosomes derived from human umbilical cord mesenchymal stromal cell. Cytotherapy. 2016 Mar;18(3):413-22. doi: 10.1016/j.jcyt.2015.11.018).
Moreover, there is evidence of the possibility of using exosomes of plant origin for the treatment of various pathological processes in mammals. Currently, there is evidence that vesicles of plant origin may be involved not only in plant–cell communication but also in interspecies communication between plants and animals. For example, a plant-derived miRNA such as miR-168 has been reported to enter the circulation of rice-fed mice enclosed in vesicles and to modulate the expression of target genes (Zhang L, Hou D, Chen X, Li D, Zhu L, Zhang Y, Li J, Bian Z, Liang X, Cai X, Yin Y, Wang C, Zhang T, Zhu D, Zhang D, Xu J, Chen Q, Ba Y, Liu J, Wang Q, Chen J, Wang J, Wang M, Zhang Q, Zhang J, Zen K, Zhang CY. Exogenous plant MIR168a specifically targets mammalian LDLRAP1: evidence of cross-kingdom regulation by microRNA. Cell Res. 2012 Jan;22(1):107-26. doi: 10.1038/cr.2011.158. Epub 2011 Sep 20. Erratum in: Cell Res. 2012 Jan;22(1):273-4. PMID: 21931358; PMCID: PMC3351925.). Exosomes are released from microorganisms and may participate in interspecies communication in the gut (Lawson C, Kovacs D, Finding E, Ulfelder E, Luis-Fuentes V. Extracellular Vesicles: Evolutionarily Conserved Mediators of Intercellular Communication. Yale J Biol Med. 2017 Sep 25;90(3):481-491. PMID: 28955186; PMCID: PMC5612190.).
Based on the above, a decision was made on the possibility of transplanting EVs obtained from rat MSCs into rabbits.
The text of the manuscript is supplemented
- Did the Authors use FBS-containing medium to collect MSC-EVs? The “Materials and methods” section does not clearly describe the procedure. In case if MSC-EVs were collected in serum-supplemented cell culture medium, additional controls should be used to prove that truly MSC-MVs (not FBS-EVs) were analyzed in the study.
The MSCs were cultured in a nutrient medium containing 7% fetal bovine serum (Biocera, Nuaille, France), which was preliminarily purified from its own EVs by complete gradient centrifugation.
The text of the manuscript is supplemented
- The information abut PKH26 dye in the “Materials and methods” section is incomplete. Although it can be concluded from the results and the discussion part that red fluorescence was used, such information should be included in the materials section.
I apologize for vaguely expressing information about PKH-26 (Sigma-Aldrich, USA). But the data on the red fluorescence of this dye is contained in the section “Materials and methods”, in the last 2 paragraphs.
The text has been left unchanged, the information required for the Reviewer is highlighted in color in the last 2 paragraphs in the section “Materials and methods”.
Minor points:
- In the introduction, the Authors cite two studies utilizing fluorescently labeled EVs (ref 11 and 12). However, MSC may originate from different tissues (bone marrow, adipose tissue, umbilical cord, etc.), which may affect EV properties. Therefore, it should be stated which population of MSCs was exactly used in these studies.
I have to draw the attention of the Reviewer that this information is contained in the section "Materials and methods" (MSCs were obtained from the bone marrow of a male Wag inbred line rat...).
The text has been left unchanged, the information required for the Reviewer is highlighted in color in the section “Materials and methods”.
- The “Author contributions” section is not completed. Please, complete.
Changed
- The “Acknowledgements” are incorrect. Please, provide correct version.
Changed
I once again want to thank the distinguished reviewer for the clear and specific recommendations and for the work spent studying our manuscript.
Changes to the text are highlighted in color.

Round 2
Reviewer 1 Report
The manuscript studied Multipotent Stromal Cells (MSCs) spread from the injection site to different organs. They obtained MSCs from cultivated rat MSCs to injected into rabbit limbs after staining them with PKH26. Phosphate buffer injection was substituted for MSCs in the control group. The luminescent mode microscope images present fluorescence objects in the lungs, heart, and spleen of the test groups after 3,7 and 10 days.
The manuscript discussed the investigation very well. The following points could be helpful.
- The following sentence was repeated in both the abstract and introduction.
“The possibility of generalized spread of MSC EVs to distant organs in case of local intratissular administration remains unexplored.”
- The following two sentences are large and complex; they can be more fluent by simple and straightforward sentences.
“In this regard, it is relevant to study the possibility of MSC EV bio-distribution throughout the body after a local administration, since it is impossible to exclude their influence on tissues distant from the injection site due to the active or passive migration of injected MSC EVs through the vessels.”
Nanovesicles labeled Cy7 (cyanine fluorescent dye), which were obtained from MSCs by isolation in a density gradient due to the destruction of cellular elements by successive extrusions, were injected intraperitoneally at a dose of 2 × 109 into mice, disseminated throughout the animal’s body, and localized in the lungs, liver, and kidneys after 6 hours.
Author Response
I am grateful to the Reviewer for studying our manuscript and for the time spent.
The following points could be helpful.
- The following sentence was repeated in both the abstract and introduction.
“The possibility of generalized spread of MSC EVs to distant organs in case of local intratissular administration remains unexplored.”
This sentence substantiates the aim of the research, informs the reason for the creation of this manuscript. Since researchers first read only abstracts, and only if they are very interested in the article, proceed to reading the manuscript itself, we repeat the rationale for doing this work both in the abstract and in the introduction to the article.
- The following two sentences are large and complex; they can be more fluent by simple and straightforward sentences.
“In this regard, it is relevant to study the possibility of MSC EV bio-distribution throughout the body after a local administration, since it is impossible to exclude their influence on tissues distant from the injection site due to the active or passive migration of injected MSC EVs through the vessels.”
fixed
Nanovesicles labeled Cy7 (cyanine fluorescent dye), which were obtained from MSCs by isolation in a density gradient due to the destruction of cellular elements by successive extrusions, were injected intraperitoneally at a dose of 2 × 109 into mice, disseminated throughout the animal’s body, and localized in the lungs, liver, and kidneys after 6 hours.
fixed
Changes to the text are highlighted in color.
I thank him again for the work done.

Reviewer 2 Report
I thank the Authors for responding to my comments and for providing explanation to my concerns. Unfortunately, I can not find any additional data on EV characterization, which is indispensable when working with EVs. Although the Authors added a description of EV analysis to the “Materials and methods”, they did not provide any supportive figure. Without showing the minimal data on EVs, the Authors can not claim that they indeed, used MSC-EVs in their work. Additionally, the Authors should follow the latest ISEV guidelines as described in the following paper: doi: 10.1080/20013078.2018.1535750, instead of earlier criteria (ref. 20, 21).
Author Response
I thank the Authors for responding to my comments and for providing explanation to my concerns. Unfortunately, I can not find any additional data on EV characterization, which is indispensable when working with EVs. Although the Authors added a description of EV analysis to the “Materials and methods”, they did not provide any supportive figure. Without showing the minimal data on EVs, the Authors can not claim that they indeed, used MSC-EVs in their work. Additionally, the Authors should follow the latest ISEV guidelines as described in the following paper: doi: 10.1080/20013078.2018.1535750, instead of earlier criteria (ref. 20, 21).
The experiment, the results of which are presented in this manuscript, was not planned and done yesterday, or even last year. At the time of planning and doing the work, the recommendations of earlier authors (ref. 20, 21) were relevant. In addition, the manuscript does not describe the results of the using EVs, but only proves the possibility of their dissemination throughout the body after being introduced not into the blood, but into the tissues. Therefore, we hope that the minimum characteristics of the applying EVs, based on the “old” recommendations, did not significantly affect the data on the distribution of these particles in different organs.
When preparing samples for research, the objects were sorbed onto a copper grid covered with a formvar film for 1 min and contrasted with a 2% phosphotungstic acid solution for 10 sec. The grids were studied in the transmission mode of a Jem1400 electron microscope (Jeol, Japan); images were obtained using a Veleta digital camera (Olympus Corporation, Japan). Particle size was determined in 5-8 randomly selected fields of view at 60 000 times magnification using the iTEM software package (Olympus Corporation, Japan). More than 90% of the objects had a diameter of 70-90 microns and a three-layer membrane.
The isolated extracellular particles were adsorbed onto 4 μm aldehyde/sulfate latex particles (Invitrogen, 37304) (Dias MV, Martins VR, Hajj GN. Stress-Inducible Protein 1 (STI1): Extracellular Vesicle Analysis and Quantification. Methods Mol Biol. 2016;1459:161-74. doi: 10.1007/978-1-4939-3804-9_11.) and analyzed on a NovoCyte ™ cytometer using its software (ACEA Biosciences Inc.). To detect EVs, antibodies specific to marker-specific exosome proteins (Bio-Rad, MCA4754F, FITC mouse IgG1, k) were used. An isotype control (Bio-Rad, MCA1209, isotype control, FITC mouse IgG1, k) was used to measure nonspecific sorption. In each experiment, no less than 30 000 events were counted.
The text of the manuscript is supplemented
Changes to the text are highlighted in color.
